# Understanding the Socio-Demographic Profile of Waste Re-Users in a Suburban Setting in South Africa

**Samantha Louise Lange** [1,*], **Mpinane Flory Senekane** [2] and **Nisha Naicker** [2]

1   Water and Health Research Centre, Faculty of Health Sciences, University of Johannesburg, Johannesburg 2028, South Africa
2   Environmental Health Department, Faculty of Health Sciences, University of Johannesburg, Johannesburg 2028, South Africa; msenekane@uj.ac.za (M.F.S.); nnaicker@uj.ac.za (N.N.)
*   Correspondence: samlange18@gmail.com; Tel.: +27-72-200-3882

**Abstract:** Minimising waste through re-use thereof is the third tier of the waste minimisation hierarchy. Understanding the socio-demographic profile of waste re-users can assist in developing effective waste minimisation strategies and programmes. The objective of this paper was to understand the socio-demographic profile of waste re-users and determine whether this affected their re-use activities and pro-environmental behaviour (PEB). This was a cross-sectional study conducted in three randomly selected suburbs in Gauteng, South Africa ($n = 150$). The data was retrieved using a questionnaire and multivariate analysis using a MAONVA test was conducted to assess which factors were associated with PEB and waste re-use. Women re-used plastic containers more than males and homeowners re-used glass jars more than tenants. The level of education had no significant bearing on specific re-use activities. Multivariate analysis results indicate that gender, age groups and type/status of homeownership played a significant role in statements that negatively influence waste re-use. Based on the results of this study, the best candidates for re-use activities and PEB in suburban communities in South Africa are women homeowners aged between 31 and 50 years.

**Keywords:** domestic waste; re-use; pro-environmental behaviour; gender; waste minimisation; socio-demographics

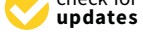



## 1. Introduction

The waste hierarchy has existed for at least forty years and its principles concentrate on prioritizing reduction in waste through minimizing waste materials, re-use and recycling of waste [1]. In South Africa, the National Waste Management Strategy seeks to provide a common platform for all stakeholders so that waste management in South Africa can be systematically improved [2]. The strategy was developed to comply with Section 6(1) of the National Environmental Management Waste Act, Waste Act 29 of 2008, which requires the development of a National Waste Management Strategy [3]. Within the strategy, the waste hierarchy is defined as having five tiers. The first tier of the hierarchy deals with waste minimization and reduction at source, with the second tier focusing on re-use and the final and least desirable outcome for waste being the fifth tier, which is disposal. The waste hierarchy has evolved over time to include waste avoidance as the first and most desirable option, moving waste re-use to the third tier ahead of recycling and recovery [4], as depicted in Figure 1.

Organic waste contributes over nineteen million tons towards the overall annual waste burden in South Africa [3]. Very little action is taken within the country to reduce this amount, however, a study in Poland showed that by using surplus rapeseed meal and microcrystalline cellulose they were able to produce a biodegradable packaging, thereby successfully reducing the amount of potential agricultural waste and turning it into a sustainable, usable product [5]. These biodegradable solutions can assist in reducing waste, although more research needs to be conducted to ensure that the environmental benefits

outweigh the negative impacts of possible increases in greenhouse gasses due to the final biodegradability at landfill sites through increased "food" waste [6] as well as whether landfill sites have the capacity and infrastructure for such innovations [7].

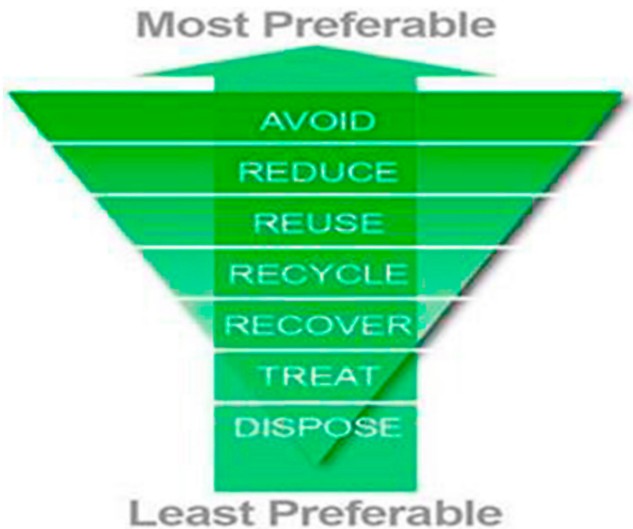

**Figure 1.** Seven-tier waste hierarchy [4].

Ensuring that materials are re-used before they become waste reduces the burden on waste managers by not only reducing the amount of waste at the landfill but also by circumventing the illegal re-use of an item which has initially been classified as waste [8].

South Africa has adopted the internationally accepted wording for the definition of re-use, which means any operation by which end-of-life products or equipment or components thereof are used for the same or similar purpose for which they are conceived [9]. The definition found in the Waste Act, 29 of 2008 concurs by defining the re-use of waste as utilising articles from a waste stream again, for a similar or different purpose, without changing the form or properties of the article [3]. Some of the more common activities aligned with waste re-use have been identified for this study, namely: the re-use of shopping bags; reusing glass jars and bottles; reusing plastic containers; reusing articles for creative purposes; repairing broken items; donating items to charity shops or domestic workers; buying or selling items at second-hand shops and alternative uses for waste such as birdfeeders and flowerpots. Reusing potential waste articles was found to be more efficient than recycling as there was less cost involved in cleaning and reusing an article or item than recycling it into a new product [10]. Internationally, commonly re-used items are plastic bottles which are re-filled with water or soft drinks [11,12] and clothing which is passed on or donated to others [13]. Re-use of primary packaging for food products is not a popular practice due to the possibility of contamination of the product, however, it is gaining popularity for secondary packaging such as reusable crates and containers for primary packaged goods [14].

Pro-environmental behaviour (PEB) can be defined as actions aimed at avoiding harm to and/or safeguarding the environment and are either found in public (participation in environmental movements) or private (recycling and re-use) domains [15]. Re-use of domestic waste falls within this behavioural sphere. Numerous studies show that age and gender can play a role regarding PEB. Data extracted from a study of 90,000 people in 40,000 households in Great Britain [16] showed that in both couples who had no children and those that did have children, as well as women that were older, were more inclined towards pro-environmental behaviour. The same study also showed that people who were of a pensionable age were comparatively less inclined towards such behaviour, whereas an Indian study concluded that males score higher on PEB, with middle-aged persons and married persons also scoring higher [17]. These results differ from a study conducted by Chen et al. [18], who stated that young, female and highly educated persons were

more pro-environmentally inclined [18]. In a study of Australian employees, 84.8% of the respondents indicated that they were concerned about environmental problems. The largest group were those aged between 30–44 and 45–64 years, followed by those aged 65+. There was significantly less concern amongst 18–29 year olds. The group most concerned with environmental problems was women aged 45–64 years and the least concerned group was males aged 18–29 years [19]. A sample of university students indicated no difference between males and females with regard to PEB and that in general their PEB was moderately high, with those in the College of Natural resources significantly higher than students from the College of Agriculture [20]. A review of the literature showed that exposing adolescents to pro-environmental behaviour, particularly amongst students who are strongly committed to raising environmental awareness, could be an important factor in changing behaviour, especially amongst a cohort who will be future policy makers [21].

A review of the literature indicates very little information regarding re-use of waste, specifically in South Africa, and indicates conflicting results on the demographics of waste re-users or persons with PEB. Researchers agree that re-use of waste is an essential and economically sound practice to reduce waste entering the waste stream [10]. With over fifty-five million tons of waste entering an overburdened, under-resourced waste disposal system in South Africa, it is essential for all areas of waste minimization to be explored so that they can be developed to their fullest potential. This study seeks to discover the demographics of waste re-users in South African households, including those most prone towards PEB and by implication, identifying those least likely to do so. This can assist decision makers, product manufacturers, retailers and any other waste-producing industries to promote this very important and accessible waste minimisation practice by ensuring that the manufacture and distribution of articles with a potential to become waste can be diverted into a re-usable alternative by targeting the demographic most likely to do so, or alternatively developing programmes to target the least likely demographics.

## 2. Materials and Methods

This is a retrospective analysis of data collected from a cross-sectional study conducted in 2014 [22].

### 2.1. Study Area

Suburbs were randomly selected through a stratified random sampling process. The suburban areas of Kempton Park, an area within the City of Ekurhuleni, Gauteng, were divided into three sections to the north, east and west of the central business district of Kempton Park. The suburbs excluded agricultural areas and industrial areas and consisted mainly of residential properties interspersed with a few commercial properties. Each of the suburbs received municipal services including kerbside refuse collection.

### 2.2. Sampling

A suburb from each of these three groups was then randomly selected. A map of each area showing the household stands was used to draw the sample. The maps which are part of the City of Ekurhuleni Geographic Information Sytem (GIS) showed street addresses and stand numbers for each stand in Kempton Park West, Bonaero Park and Glen Marais to enable identification. The sample areas are indicated in Figure 2.

Only one household per stand was selected. Households were described as a group of people who jointly provide themselves with food and other living essentials. It may also be a single individual living alone [15]. Any person within a selected household that was over the age of 18 years that agreed to complete the questionnaire became the respondent. Only one household member per household participated. A household member is described as someone who resides in the household for at least four nights of the week [23]. A sample-size calculator required a sample size of 383. Due to logistical and time constraints, it was decided to reduce the sample to 50 households per area to provide a sample of 150 households, which equated to a 95% confidence level with an 8% margin of error,

similar to a study conducted in the Northern Cape, South Africa [13]. The reduction in sample size would further be justified by ensuring a 100% response rate.

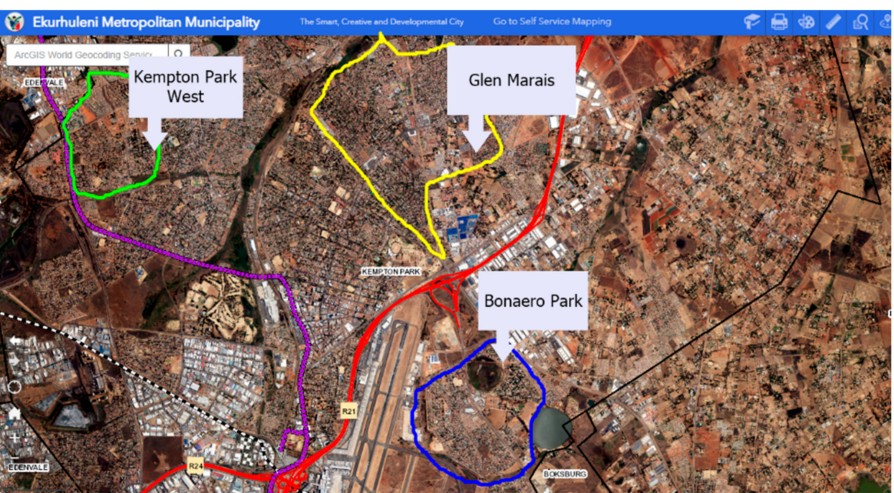

**Figure 2.** Location of sample areas in Kempton Park.

*2.3. Data Collection Tool*

The questionnaire was divided into four sections. Section one dealt with the demographic profile of respondents; section two covered the types of re-use activities respondents engaged in; section three referred to where respondents obtained information on waste re-use and section four dealt with their PEB and actions. This article examines the results of sections one, two and four. Section one included age, gender, level of education and residential status of respondents. The residential status indicated whether the respondent was the homeowner or the tenant of the premises in question. The second section of the questionnaire requested information on domestic waste re-use activities. Respondents were required to tick one or more of the 10 commonly aligned domestic waste re-use activities that they participated in. There was also an opportunity for respondents to report no re-use activity and other. The final section of the questionnaire comprised a set of questions addressing PEB derived from a manual designed to assist health researchers in drawing up such questionnaires [24] examining factors which influence re-use behaviour.

Questionnaires were hand delivered door-to-door by the researcher to the selected households for self-administration. Households were not informed of the study prior to the data collection period, however, each respondent received and information letter at the time of collection and was required to give informed consent before proceeding. In most cases, the respondent did not wish to complete the questionnaire on their own and the researchers were available to administer the questionnaire and explain any statements which were unclear to respondents. The administration of the questionnaires took place during May and June 2014. Each questionnaire took approximately 20 min to administer. Respondents were mostly, but not limited to, the person most responsible for managing domestic waste. When a selected household did not wish to participate in the survey the specific sample was noted as non-responsive. In the case of a non-responsive household, the households on either side of the selected household were approached. If they were also non-responsive, an additional household was selected by making use of the simple random sampling method described above. This was performed to ensure the minimum amount of 50 samples per suburb to reach the target of 150 samples.

*2.4. Statistical Analysis*

Data was analysed using IBM SPSS (Statistical Software for Social Sciences) Version 28 to determine frequencies and means. A Pearson's chi-square analysis was used to analyse significance of results where common re-use activities were cross-tabulated with the demographics of the study participants ($p < 0.05$). A similar analysis was conducted

comparing demographics with several PEB questions. A 5-point Likert scale was used where 1= strongly agree and 5 =strongly disagree. The frequency and mean of Likert-scale questions were determined and reported. There were seventeen questions placed into three groups, namely: exclusivity of re-use practisers; positive influences of re-use practices and negative influences. The groups were tested for reliability using Cronbach's alpha ($\alpha > 0.7$) and the combined means were reported. Each of these groups of questions was exposed to a multivariate test (multivariate analysis of variance—MANOVA) against the various socio-demographic variables using Wilk's lambda for significance ($p = 0.05$).

## 3. Results

The results indicate that 46.7% of respondents were male and 53.3% were female, with 40.0% of these being in the 31–50 years age group. Figure 3 indicates the age distribution of respondents. Just over half of the respondents (53.3%) possessed a secondary school education and a further 43.4% had a tertiary or post-graduate qualification, as seen in Figure 4.

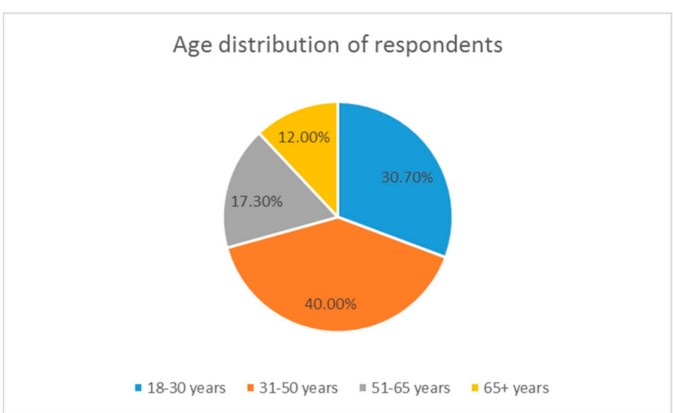

**Figure 3.** Age distribution of respondents.

Homeowners made up 56.0% of the respondents, with 66.0% indicating that they were responsible for the disposal of household refuse, meaning that they ensured that refuse was placed kerbside for collection or otherwise disposed of. Most respondents participated in some re-use activity, although nine (6.0%) indicated that they did not practice any form of re-use activity. There was no socio-demographic profile which was found to be significantly different among any of the respondents.

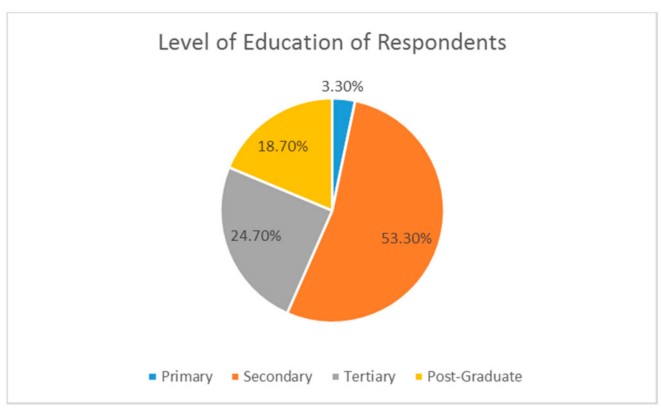

**Figure 4.** Level of education of respondents.

### 3.1. Re-Use Activities

Table 1 indicates the frequency of common waste re-use practices and was reported according to the various demographic variables.

**Table 1.** Socio-demographic profile of participants and their re-use activities.

| | | Respondents | Shopping Bags | Glass Jars | Plastic Containers | Repair Broken Items | Creative Projects | No Re-Use | Donating Items to Charity | Donating Items to Domestic | Selling Items | Buying Second-Hand | Alternate Uses | Other |
|---|---|---|---|---|---|---|---|---|---|---|---|---|---|---|
| *n* = 150 | | | 125 (83.3%) | 97 (64.7%) | 100 (66.7%) | 56 (37.3%) | 57 (38.0%) | 9 (6.0%) | 105 (70.0%) | 99 (66.0%) | 30 (20.0%) | 52 (34.7%) | 39 (26.0%) | 1 (0.7%) |
| Gender | Male | 70 (46.7%) | 56 (80.0%) | 40 (57.1%) | 38 (54.3%) | 30 (42.8%) | 21 (30.0%) | 5 (7.0%) | 45 (64.2%) | 42 (60.0%) | 16 (22.8%) | 31 (44.3%) | 18 25.7%) | 0 (0.0%) |
| | Female | 80 (53.3%) | 69 (86.2%) | 57 (71.2%) | 62 (77.5%) | 26 32.5%) | 36 (78.7%) | 4 (5.0%) | 60 (75.0%) | 57 (71.2%) | 14 (17.5%) | 21 (26.2%) | 21 (26.2%) | 1 (1.2%) |
| *p* Value | | | 0.306 | 0.710 | 0.003 | 0.191 | 0.059 | 0.581 | 0.153 | 0.147 | 0.413 | 0.021 | 0.941 | 0.348 |
| Age Group | 18–30 years | 46 (30.7%) | 35 (76.0%) | 29 (63.0%) | 29 (63.0%) | 17 (37.0%) | 17 (37.0%) | 4 (8.7%) | 29 (63.0%) | 23 (50.0%) | 12 (26.1%) | 22 (47.8%) | 11 (23.9%) | 1 (2.1%) |
| | 31–50 years | 60 (40.0%) | 52 (86.7%) | 36 (60.0%) | 43 (71.7%) | 24 (40.0%) | 24 (40.0%) | 3 (5.0%) | 42 (70.0%) | 45 (75.0%) | 15 (25.0%) | 22 (36.7%) | 13 (21.7%) | 0 (0.0%) |
| | 51–65 years | 26 (17.3%) | 23 (88.5%) | 20 (76.9%) | 17 (65.4%) | 7 (26.9%) | 11 (42.3%) | 1 (3.8%) | 20 (76.9%) | 20 (76.9%) | 1 (3.8%) | 3 (11.5%) | 8 (30.8%) | 0 (0.0%) |
| | >65 years | 18 (12.0%) | 15 (83.3%) | 12 (66.7%) | 11 (61.1%) | 8 (44.4%) | 5 (27.8%) | 1 (5.6%) | 14 (77.8%) | 11 (61.1%) | 2 (11.1%) | 5 (27.8%) | 7 (38.9%) | 0 (0.0%) |
| *p* Value | | | 0.438 | 0.500 | 0.749 | 0.620 | 0.771 | 0.821 | 0.537 | 0.029 | 0.068 | 0.017 | 0.466 | 0.517 |
| Level of Education | Primary | 5 (3.3%) | 4 (80.0%) | 2 (40.0%) | 2 (40.0%) | 1 (20.0%) | 2 (40.0%) | 0 (0.0%) | 2 (40.0%) | 1 (20.0%) | 2 (40.0%) | 2 (20.0%) | 2 (20.0%) | 0 (0.0%) |
| | Secondary | 80 (53.3%) | 65 (81.2%) | 55 (68.7%) | 57 (71.2%) | 30 (37.5%) | 24 (30.0%) | 7 (8.7%) | 57 (71.2%) | 54 (67.5%) | 13 (16.2%) | 32 (40.0%) | 19 (23.7%) | 1 (1.2%) |
| | Tertiary | 37 (24.7%) | 30 (81.1%) | 23 (62.2%) | 20 (54.0%) | 14 (37.8%) | 16 (43.2%) | 1 (2.7%) | 25 (67.6%) | 25 (67.6%) | 9 ;(24.3%) | 11 (29.7%) | 11 (29.7%) | 0 (0.0%) |
| | Post-Graduate | 28 (18.7%) | 26 (92.8%) | 17 (60.7%) | 21 (75.0%) | 11 (39.3%) | 15 (53.6%) | 1 (3.5%) | 21 (75.0%) | 19 (67.8%) | 6 (21.4%) | 7 (25.0%) | 7 (25.0%) | 0 (0.0%) |
| *p* Value | | | 0.521 | 0.530 | 0.118 | 0.875 | 0.139 | 0.494 | 0.451 | 0.181 | 0.490 | 0.454 | 0.801 | 0.830 |
| Domestic Responsibility | Myself | 99 (66.0%) | 83 (83.8%) | 67 (67.7%) | 66 (66.7%) | 40 (40.4%) | 37 (37.4%) | 5 (5.0%) | 68 (68.7%) | 60 (60.6%) | 17 (17.2%) | 33 (33.3%) | 26 (26.3%) | 1 (1.0%) |
| | My spouse | 11 (7.3%) | 9 (81.8%) | 5 (55.5%) | 6 (54.5%) | 3 (27.3%) | 3 (27.3%) | 3 (27.3%) | 7 (63.6%) | 8 (72.7%) | 3 (27.3%) | 2 (18.2%) | 5 (45.4%) | 0 (0.0%) |
| | My Children | 3 (2.0%) | 3 (100.0%) | 2 (66.7%) | 2 (66.7%) | 2 (66.7%) | 2 (66.7%) | 0 (0.0%) | 2 (66.7%) | 3 (100.0%) | 1 (33.3%) | 1 (33.3%) | 1 (33.3%) | 0 (0.0%) |
| | My domestic worker | 24 (16.0%) | 19 (79.2%) | 12 (50.0%) | 18 (75.0%) | 10 (41.7%) | 10 (41.7%) | 1 (4.2%) | 17 70.8%) | 20 (83.3%) | 6 (25.0%) | 8 (33.3%) | 3 (12.5%) | 0 (0.0%) |
| | Other | 13 (8.7%) | 11 (84.6%) | 11 (84.6%) | 8 (61.5%) | 1 (7.7%) | 5 (38.5%) | 0 (0.0%) | 11 (84.6%0 | 8 (61.5%) | 3 (23.0%) | 8 (61.5%) | 4 (30.8%) | 0 (0.0%) |
| *p* Value | | | 0.917 | 0.153 | 0.803 | 0.133 | 0.784 | 0.038 | 0.802 | 0.173 | 0.801 | 0.234 | 0.322 | 0.972 |
| Residential status | No Response | 4 (2.7%) | 3 (75.0%) | 1 (25.0%) | 2 (50.0%) | 2 (50.0%) | 3 (75.0%) | 0 (0.0%) | 4 (100.0%) | 3 (75.0%) | 1 (25.0%) | 1 (25.0%) | 0 (0.0%) | 0 (0.0%) |
| | Homeowner | 84 (56.0%) | 75 (89.3%) | 62 (73.8%) | 63 (75.0%) | 29 (34.5%) | 36 (42.8%) | 4 (4.8%) | 84 (100.0%) | 66 (78.6%) | 15 (17.8%) | 25 (29.8%) | 22 (26.2%) | 0 (0.0%) |
| | Tenant | 61 (40.7%) | 46 (75.4%) | 33 (54.1%) | 34 (55.7%) | 25 (41.0%) | 18 (29.5%) | 5 (8.1%) | 61 (100.0%) | 30 (49.2%) | 14 (23.0%) | 26 (42.6%) | 17 (27.9%) | 1 (1.6%) |
| | Other | 1 (0.7%) | 1 (100%) | 1 (100%) | 1 (100%) | 0 (0.0%) | 0 (0.0%) | 0 (0.0%) | 1 (100%) | 0 (0.0%) | 0 (0.0%) | 0 (0.0%) | 0 (0.0%) | 0 (0.0%) |
| *p* Value | | | 0.151 | 0.025 | 0.075 | 0.682 | 0.130 | 0.784 | 0.009 | 0.001 | 0.829 | 0.349 | 0.600 | 0.689 |

The re-use of plastic containers was significantly more prevalent amongst females, with 77.5% female respondents participating in this activity (*p* = 0.003). Conversely, the buying of second-hand goods is significantly more frequent amongst male participants (*p* = 0.021). Age was a factor in two re-use activities, namely: donating unwanted items to household domestic workers and buying second-hand goods. The donating of items to household domestic workers occurred more frequently in the 31–50-year age category, with 75% of persons donating (*p* = 0.029). Respondents in both the 18–31 and 31–50-year age group were both found to buy goods from second-hand stores, with the 51–65 and above 65-year age groups being significantly lower (*p* = 0.017). The level of education of the respondents had no significant impact of their re-use activities.

Respondents indicated who was responsible for placing household refuse kerbside for collection. Interestingly, this had no significant impact on re-use activities, however, respondents indicating no re-use activity, but being responsible for household waste disposal, was significant (*p* = 0.038). Being a homeowner played a significant role in the practice of reusing glass jars and bottles (*p* = 0.025), donating items to charity (*p* = 0.009) and donating unwanted items to household domestic workers (*p* = 0.001).

### 3.2. Influences of Re-Use Behaviour

A section of the questionnaire comprised of PEB-based questions and means were calculated for each question from the 5-point Likert scale. There were twenty-five statements of behaviours and ideas which could influence re-use activities and subsequently PEB. Seventeen of these statements, which were consistently reliable, were linked together to provide a common mean for three influences, namely exclusivity of re-use practices by individuals, positive influences and negative influences.

Table 2 indicates the combined statements and their joint means as well as the individual means (1 = strongly agree and 5 = strongly disagree) for each question.

**Table 2.** Statements to determine influences of re-use behaviour.

| Statement | % | +Mean | SD | α * | % | Mean ** | SD *** |
|---|---|---|---|---|---|---|---|
| Re-use: Exclusivity of practice | | | | 0.877 | 85 | 21.21 | 3.74 |
| Only children under 18 years of age should participate in reusing domestic waste | 82 | 4.12 | 1.00 | | | | |
| Only females should participate in reusing domestic waste | 87 | 4.35 | 0.83 | | | | |
| Only males should participate in reusing domestic waste | 88 | 4.39 | 0.79 | | | | |
| Only environmentally aware people should re-use waste | 82 | 4.10 | 1.03 | | | | |
| Only municipal employees should participate in reusing domestic waste | 85 | 4.24 | 0.89 | | | | |
| Re-use: Positive influences | | | | 0.838 | 60 | 14.89 | 4.99 |
| Reusing domestic waste makes me feel useful | 42 | 2.12 | 0.98 | | | | |
| Reusing domestic waste will help future generations | 31 | 1.57 | 0.84 | | | | |
| Reusing domestic waste reduces the amount of waste I throw away | 35 | 1.75 | 0.90 | | | | |
| Reusing domestic waste is a convenient activity for me | 50 | 2.49 | 1.09 | | | | |
| Reusing domestic waste keeps our brains active by thinking about creativity | 44 | 2.21 | 1.08 | | | | |
| Reusing domestic waste reduces waste to the landfill site | 33 | 1.65 | 0.78 | | | | |
| Reusing domestic waste keeps our environment clean and healthy | 31 | 1.54 | 0.79 | | | | |
| I feel reusing domestic waste is a positive activity | 31 | 1.55 | 0.78 | | | | |
| Re-use: Negative influences | | | | 0.749 | 56 | 14.18 | 3.45 |
| Reusing domestic waste can sometimes be expensive | 75 | 3.77 | 1.04 | | | | |
| Reusing domestic waste is time consuming | 67 | 3.37 | 1.24 | | | | |
| Reusing domestic waste is unhygienic | | 3.39 | 1.18 | | | | |
| Reusing domestic waste is harmful to me | | 3.65 | 1.09 | | | | |

+Mean of Likert-scale where 1 = strongly agree and 5 = strongly disagree; * Cronbach's alpha for reliability of statements; ** combined mean of statements; *** combined SD.

The percentage of each mean is indicated with a result of <50% indicating the mean leaning more towards strongly agreeing with the statement and >50% indicating strongly

disagreeing with the statement. The maximum combined mean score to strongly disagree would be 25, with a maximum mean score to strongly agree with the statements being 5. A score of 12.5 indicates neither agreeing nor disagreeing with the statement. Regarding the exclusivity of the practice of re-use of domestic waste, the respondents disagreed with the statements that only certain people should participate in the practice (combined mean = 21.21; SD 3.74). This equates to there being an 85% disagreement with the statement. In statements providing positive and negative motivations, the respondents reported a mean of 14.89 (SD 4.99) for positive (60% disagreement with the positive reuse statements) and 14.18 (SD 3.45) for negative motivations, providing a 56% disagreement with negative influences such as re-use being unhygienic and time consuming.

Results of the multivariate test (MANOVA) of statements dealing with exclusivity of re-use practices and positive and negative influences of waste re-use are captured in Table 3. The multivariate test showed that residential status was significant in disagreeing with the statements of who should exclusively re-use waste ($p = 0.003$). Gender played a significant role in agreeing or disagreeing with statements that negatively influence waste re-use ($p = 0.036$). Results indicate that age groups also played a role in the negative statements ($p = 0.012$), as well as residential status ($p = 0.000$).

**Table 3.** Statements regarding influences of waste re-use (MANOVA).

| | Exclusivity of Practice | | Positive Influences | | Negative Influences | |
|---|---|---|---|---|---|---|
| | Wilks' Lambda | Sig. | Wilks' Lambda | Sig. | Wilks' Lambda | Sig. |
| Gender | 0.961 | 0.328 | 0.971 | 0.834 | 0.932 | 0.036 |
| Age group | 0.903 | 0.470 | 0.850 | 0.053 | 0.837 | 0.012 |
| Level of education | 0.899 | 0.321 | 0.803 | 0.145 | 0.903 | 0.252 |
| Domestic responsibility | 0.930 | 0.960 | 0.747 | 0.121 | 0.903 | 0.540 |
| Residential status | 0.786 | 0.003 | 0.789 | 0.088 | 0.761 | 0.000 |

## 4. Discussion

There is no doubt that the re-use of waste is a valuable tool in reducing the amount of waste disposed of at landfill sites. The importance of preserving the sustainability of landfills in order to reduce the unnecessary use of land for such purposes cannot be over-emphasized. However, a literature review of factors which influence people's PEB and their re-use activities has proven contradictory.

The reuse of plastic shopping bags is the most prevalant reuse activity, with 83.3% of respondents acknowledging that they did so. This could be a habitual practice as there have been several drives by retailers to encourage consumers to move away from single-use shopping bags. In addition, South Africa and its neighbouring countries of Botswana and Zimbabwe have imposed a levy on plastic bags since 2003 [14]. The results of this study show that certain demographics are more prone to certain activities and behaviours or thoughts. The re-use of plastics is significantly higher amongst females ($p = 0.003$), which could be attributed to the traditional role of females in domestic settings where females prepare food and dispose of or re-use containers which lend themselves to this purpose. This result is reinforced by studies which show that household PEB and related activities occurred when females were involved in traditional or typical roles [8]. In this study, buying second-hand goods was a predominantly male activity ($p = 0.021$), which is in opposition to a Polish study agreeing with previous studies that women are more likely to buy second-hand goods, although only 27% of participants were male [25] compared with this study with a more evenly distributed gender demographic.

The donating of goods to domestic workers was significant amongst the 31–50 year-aged group ($p = 0.029$) and homeowners ($p = 0.001$). In South Africa, there are approximately 892,000 domestic workers employed in households, with the largest age group of employed persons aged 35–44 years [26], which would infer the significance of this age group being employed and, therefore, employing a domestic worker to assist with running the

household. A household study in Northern Cape, South Africa showed that one of the few re-use activities practiced by households was the donation of used clothes, with 32.7% of households participating in this [13].

Homeowners are also more likely to be employed or more financially stable than tenants and, therefore, more likely to employ and donate goods to domestic workers. Homeowners also donate significantly more to charities ($p = 0.009$), possibly also due to better financial stability. Higher income groups are more likely to practice waste reduction at source through the 3R's (reduce, recycle and reuse), whereas middle-income groups tend to have discussions of the practice but do not implement it, and the lower-income groups have no organized structures and little awareness of the practices [11]. A similar observation was made in an earlier study of 46 households in Tshwane, South Africa [27].

Several studies report certain demographics being more prone to PEB such as young people or females [15,20]. The results of the multivariate analysis show that the residential status of the respondent plays a role. Homeowners disagreed significantly with the statements indicating that only certain people could re-use waste, thereby implying that all could re-use waste. The mean for negative influences of waste re-use by all participants was 14.18 (SD 3.45), indicating that just over half the participants disagreed with these statements, and significantly so depending on their residential status, gender and age. This is similar to the findings of a study in Indonesia which concluded that homeownership strengthened social awareness and an inclination towards a zero-waste environment [28].

Looking at the demographics, this could mean that more females who are aged 31–50 years and own their homes disagree with the negative statements that the re-use of waste can be a time consuming or unhygienic activity. This result is similar to other studies stating that women are more prone to PEB due to their awareness of causes and consequences of environmental harm [16,18].

Interestingly, the level of education did not play significant role in determining the re-use activity or the PEB of the respondents of this study. This deviates from findings in previous studies, which attributed increased levels of education to improved waste management practices or pro-environmental awareness [11,28,29].

This study addresses the type of person who would re-use waste in a domestic, suburban setting in South Africa. There are no known studies similar to this in the country and, therefore, the results of this study add to the body of knowledge and assist in further research on this topic. The study was restricted to 150 participants due to logistical constraints, and the results were extracted from a broader study: *The Behaviour and Attitudes of Kempton Park Communities with Reference to Reusing Domestic Waste* [22], which did not take into account the recent developments of on-line buying and selling on various internet platforms such as eBay. Additional investigation of the on-line market could indicate an improved circular economy with unwanted items recirculating through such online platforms.

## 5. Conclusions

The study revealed that demographics play a role in the type of waste re-used as well as the inclination towards pro-environmental behaviour and thinking. Traditional gender roles such as those of women being responsible for preparing food and, therefore, disposing or reusing packaging could provide an explanation for some of the results. The fact that residential status plays a role in re-use activities and PEB could be attributed to a better financial and social standing as well as the possibility that homeownership implies stability, forward thinking and a concern for a sustainable future. Based on the results of this study, the best candidate for re-use activities in suburban communities in South Africa could are women who own homes and are aged between 31 and 50 years.

Additional studies are needed to determine the exact nature of these waste re-users and whether these demographics are similar in other areas or if there is a difference between rural and urban waste re-use and PEB. There can be no doubt that developing programmes to strengthen the waste re-use activities among those groups already pre-disposed towards

positive environmental activities, as well as concentrating on those demographic groups who have shown to be ambivalent or negative towards re-use and other waste minimisation activities, would essentially improve waste minimisation strategies and decrease the burden on waste disposal systems in the country.

**Author Contributions:** Conceptualization, S.L.L. and M.F.S.; methodology, S.L.L. and M.F.S.; software, S.L.L.; validation, S.L.L. and N.N.; formal analysis, S.L.L.; investigation, S.L.L.; resources, S.L.L.; data curation, S.L.L.; writing—original draft preparation, S.L.L.; writing—review and editing, S.L.L., M.F.S. and N.N.; visualization, S.L.L. and N.N.; supervision, M.F.S. and N.N.; project administration, S.L.L.; funding acquisition, none. All authors have read and agreed to the published version of the manuscript.

**Funding:** This research received no external funding.

**Institutional Review Board Statement:** Permission was obtained from the University of Johannesburg Higher Degree Committee (HDC01-48-2014) and the Gauteng Department of Health Research Committee (31/03/2014-1) to conduct the research.

**Informed Consent Statement:** Written informed consent has been obtained from the respondents to publish this paper.

**Data Availability Statement:** Dataset available on request from the corresponding author.

**Conflicts of Interest:** The authors declare no conflict of interest.

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
