# Peer review of "Understanding the Socio-Demographic Profile of Waste Re-Users in a Suburban Setting in South Africa"

_resources, doi:10.3390/resources11050045_

Round 1

Reviewer 1 Report

The paper is interesting as authors analize people behaviuor on waste re-use by means of a survey. But, in anycase, some points must be clarified:

Page 3, Line 34. You talk about a map showing the addresses, where is that map? Can you show it? I think it could be more interesting than figure 1

Page 3, Line 44. More infomation about sample size calculation must be provided. If you say that the sample size must be 383, why do you reduce it? How do you reduce it? Is there any mathematic justification?

Page 3, Line 49. Authors say that the questionnaire was divided into four sections but only 3 were analysed. Please, indicate the content of each section.

Page 4, Line 11.  Where the selected homes previously informed about the questionnaire?

Tables 1 and 2 are not really the best way to show the information. Some pie charts should be added.

In the Conclusions section, line 26, authors say “Based on the results of this study the best candidate for re-use activities is women who own homes and are aged between28 31 and 50 years” You can’t generalize. You have to specify that in this part of South Africa this is what it is happening now.

Reviewer 2 Report

Dear Authors,

Basically, it is an interesting manuscript that covers an important topic of people's ecological behavior. The manuscript also has a scientific potential as the results are supported by statistical analysis. My doubts, however, are raised by the description of the test results. There are few quantitative values ​​in the description itself. It needs to be corrected.

Detailed comments are provided below:

Summary: It is not necessary to present the data in terms of significance levels in the abstract. You only need a general level.

Introduction, Line 25/2: Somewhere in the introduction you should also refer to other ways of reducing the amount of waste in landfills. Other than recycling. Currently, a lot of research concerns the production of modern biodegradable packaging - made of innovative biocomposites. For their production, various by-materials of agri-food processing are used, for example. I think it is worth mentioning. Check out the articles below: "The rape pomace and microcrystalline cellulose composites made by press processing". "Food residue to reinforce recycled plastic biocomposites".
You will also draw attention to a wider problem, which is the use of unnecessary biodegradable waste.
It will surely be an interesting supplement to your introduction.

Line 20/3: At the end of the introduction, you should add the justification for undertaking your research. Answer the questions: why are you doing your research, why is it so important, and what will it contribute to the development of this field of science.
Only then should you formulate the purpose of your research. You have to correct it.

Methodology, Line 26/4: Add more information about the SPSS software. Number, year, version: manufacturer, city, country. Review the entire methodology in this regard.

Results: When describing research results, you should rather use percentages (quantitative for individual behaviors). Only then can you indicate the obtained levels of statistical significance (when indicating specific values). In my opinion, it should definitely be improved. Review the entire description of the test results for this.

Support the discussion of the results with more citations of scientific publications. This also applies to the introduction. Only 18 scientific publications are not enough. 

Conclusions: You add some more perspective conclusion. A proposal that looks more forward.

Round 2

Reviewer 1 Report

Thank you for taking into account my comments.

Good luck and success!

Author Response

We appreciate your comments and revisions - thank you.

Reviewer 2 Report

Dear Authors,

After the corrections, the manuscript is better. I noticed that in references some publications are missing pages of articles (no. 5, 13, 20, 37 and others). Look at the literature for this. In addition, you should refine the references to the journal's requirements. Besides, I accept all corrections.

Author Response

We have refined the references as suggested and included page numbers where available - thank you for pointing this out.